# The Influence of Place Attachment on Heritage Discourse in Contemporary Places: A Case Study of Jordanian Byzantine Mosaics

**Hanan Jazaa Abukarki** [1,*] **, Havva Arslangazi Uzunahmet** [2] **and Zeynep Onur** [3]

1   Department of Interior Architecture, Faculty of Architecture, Near East University, Nicosia 99138, Cyprus
2   Department of Architecture, Faculty of Architecture, Near East University, Nicosia 99138, Cyprus;
    havva.arslangazi@neu.edu.tr
3   Department of Architecture, Faculty of Architecture and Fine, International Final University,
    Girne 99320, Cyprus; zeynep.onur@final.edu.tr
*   Correspondence: hananabukarki@yahoo.com

**Abstract:** This study looks at the influence of place attachment on heritage discourse in contemporary places, specifically, Byzantine mosaics in Jordan, where the Byzantine Empire left behind a rich mosaic heritage. Today, these mosaics are replicated in contemporary architectural environments. The purpose of this article is to explore the heritage discourse surrounding Byzantine mosaics in Jordanian contemporary places with a particular focus on the replication and interpretation of the mosaics in contemporary places. To do that, the relationship between the attachments to the Byzantine mosaic places, the community's level of awareness, and the replicated heritage discourse was examined. The research was conducted using several different methods, where structural equation modelling (SEM) was used to examine the relationship between these factors. Confirmatory factor analysis (CFA) was used to assess the measurement models of the latent components and examine their construct validity and reliability. In addition, the study was conducted in Madaba Archaeological Park in the Jordanian city of Madaba, known as the "city of mosaics", which is rich in tourism and culture, as a subject of study. The results show that the phenomena of heritage replications in contemporary places rely on the expanding interest in history, which is manifested through realizing the historical value and unique features of heritage. This attachment, knowledge, and understanding of heritage sites based on socio-cultural norms help shape the discourse of heritage replication in the contemporary built environment. These findings provide an understanding of the reasons behind the replication behavior of heritage designs in contemporary places, which can be supported in future research and used to create an appropriate contemporary sense of place. In addition to the possibility of using it as a strategy for the sustainability of heritage designs in local culture and contemporary places, not only in Jordan but also in other heritage environments, finally, some useful suggestions emerge on which future research can be based.

**Keywords:** place attachment; Byzantine mosaic; heritage replication; heritage discourse; contemporary places; Jordan

## 1. Introduction

The Byzantine mosaics of Jordan are among the most outstanding relics of the early Christian era in the Middle East. These exquisite mosaics have been the subject of extensive heritage discourse in recent years, with Madaba, Um er-Rasas, and Jerash among the sites where they have been discovered. In addition to their historical and aesthetic significance, these mosaics also hold great cultural and emotive significance for the Jordanian local populace.

This paper investigates the impact of place attachment on the discourse surrounding the Byzantine mosaics of Jordan. Place attachment refers to the emotive association that

individuals or communities develop with a particular location, frequently as a result of their personal experiences, recollections, and cultural heritage. In the case of the Jordanian Byzantine mosaics, place attachment is intrinsically tied to the religious and cultural identity of the local Jordanian population, which has been moulded over centuries by Christian and Islamic traditions.

The research is based on multiple sources, including a review of the relevant literature, which emphasizes the cultural and historical significance of the Byzantine mosaics in Jordan. The theoretical model examines the concept of place attachment and its influence on heritage discourse, while the method provides insights into the current state of heritage conservation in Jordan. The discussion delves into the social and cultural context surrounding the Byzantine mosaics in Jordan, examining the role of tradition and identity in moulding the relationship of the local population with these artefacts.

Previous research has shown that constructs such as sense of place, attachment to place, heritage management and practice, and heritage design iteration are well documented, but research examining the relationships between these constructs is rare in the literature. In conclusion, this study attempts to address this knowledge gap by further investigating the causal relationship between the perception of mosaic heritage places and the "replication of mosaic heritage design" in contemporary places such as public and private spaces. To achieve this aim, this study developed hypotheses based on the literature and proposed a research model to test and validate the selection of "Impact of Attachment to Mosaic Heritage places on Discourses Replicatng Mosaic Heritage". The study was also conducted in the Jordanian city of Madaba, known as the "city of mosaics", as a subject of study. The findings and results provide an understanding of the reasons behind the replication behaviour of heritage designs in contemporary places, which can be supported and used to create the appropriate sense of contemporary places. In addition to the potential of a strategy for the sustainable development of local heritage, some useful propositions have finally emerged on which future research can be based.

## 2. Literature Review

In Jordanian society, it can be observed that the Jordanian people have intense sensitivity to mosaics heritage, whereas the Byzantine mosaic sites are prevalent in different regions in Jordan [1]. However, where classical mosaics are not limited to heritage sites, it can be observed that the 'replicating mosaic heritage' appears in several contemporary places regardless of the type of place (private and public places). There is not a lot of information available on how emotive relationships to heritage places influence the presence of the 'replication of heritage designs' in a contemporary place. The research addresses this issue by examining how heritage places can provide a sense of place (place attachment) in local society, which might have a significant impact on the behaviour of placing 'replication heritage designs' into contemporary places. As a result, this research tries to investigate the place by employing testable hypotheses developed from place theory and taking into account connections that happen between concepts and dimensions. To achieve this, a collaboration between different disciplines takes place, which agrees that "understanding place in its true complexity is a multidisciplinary exercise" [2]. The simple definition of "place" is a geographical area that has human significance [3], but despite the simplicity of its definition, it is multidisciplinary. The notion of "sense of place" was put forth by human geographers in 1970 [4]; this was followed by a stream of research in social and environmental psychology that attempted to explain the associated meaning (called "sense of place"), focusing on connections to places such as families, communities, cities and natural landscapes, and the resulting connections between individuals and those places [5–7].

In general, sense of place is the meaning attached to a spatial setting by humans. It can also be called "sense of place, considered as a combination of social constructions and the interaction of the physical environment, consisting of both tangible and intangible dimensions that arise from the relational interactions between people and places" [8]. A place attachment is an emotional connection between a person and their

immediate physical surroundings or place [9]; this concept originates from attachment theory [10,11]. Numerous research studies have made efforts to comprehend the process of how inhabitants of an area develop emotional connections to the places they live in within a tourist environment. Some of the studies which have been conducted in this area include research conducted by [12–17]. Despite the diversity in the areas of focus, some studies have also given attention to the context of heritage tourism, for instance, the research conducted by [18–21]. There is a considerable body of research examining the concept of heritage, including works by [22–27]. These studies suggest that heritage plays a vital role in forming a sense of place and negotiating social, cultural, and community identity. When heritage is utilized as a cultural tool for self-expression and identity building on both individual and societal levels, it becomes the focus of heritage discourse, a topic of significant interest for researchers. Scholarly studies have emphasized the value of community involvement in a variety of facets of heritage management, such as interpretation, conservation, and tourism, as well as the artefact market. This involvement has also been linked to community outreach and social and cultural inclusivity [27–30]. In the context of both reconstructed heritage spaces and the replica artefact market, many studies have sought to explore the methods by which replicas gain value, authenticity, significance, and cultural biographies [31–34].

### 2.1. Mosaic

2.1.1. Brief History of Mosaic

Humans have historically participated in designs created by assembling small pieces of stone, tile, glass, etc. [35]. Mosaic is one of the oldest expressive arts, characterized by durability and long-term persistence [36]. The book La Mosaïque Antique [36] defines a mosaic as: "an assembly of small individual units with the aid of cement intended to form a flat or curved surface" (p. 9). Some examples of well-used design materials are shellfish, pearls, stones, ceramics, glass, etc.

The Cone Mosaic Columns from the Temple of Eanna at Uruk, 3000 BC, belong to the Sumerian civilization (see, for instance, [37]). However, evidence indicates that the earliest pebble mosaics appeared in Crete during the Neolithic era [38]. The use of mosaic for creating a meaningful sense of place has been traced back to the Greek empire during the fourth century B.C. in an ancient city called Pella. This was when the mosaic started to gain its place in history, but its usage and popularity did not increase till the second century B.C., with the rise of the Roman Empire. In the Byzantine Empire, it was adopted as a realistic way of expressing religious beliefs and indicating spiritual and divine presence within a designated portion of the society [39].

2.1.2. Jordanian Mosaic

In Jordan, numerous mosaic pavements are preserved today (Figure 1). The earliest Jordanian mosaic panel, which dates to the first century AD, was found in Mukawir Castle (Figure 2). It is displayed on the front facade of the entrances of Madaba's Archeological Park [40].

During the Byzantine period (324–636), several Churches were built. Mosaics art reached its peak during the Byzantine era. Mosaic art was used more dominantly for decoration during the Byzantine era. It was used to decorate churches, monasteries, and public meeting places with stories, maps, texts, and murals [41].

The decoration of the mosaic pavement consists of living elements, including people, animals, birds and plants, leaves, branches, grapes, and pomegranates. In addition, the mosaic pavement of the church is characterized by Greek mythology, such as the stories of Aphrodite and Adonis. These scenes are important historical documents of the daily life of members of Byzantine society, especially since records documenting daily life are scarce [42]. Whereas Byzantine mosaic sites are prevalent in different regions in Jordan, the most significant sites are in Madaba [43]. This has the most famous rarely seen mosaic pavements [44].

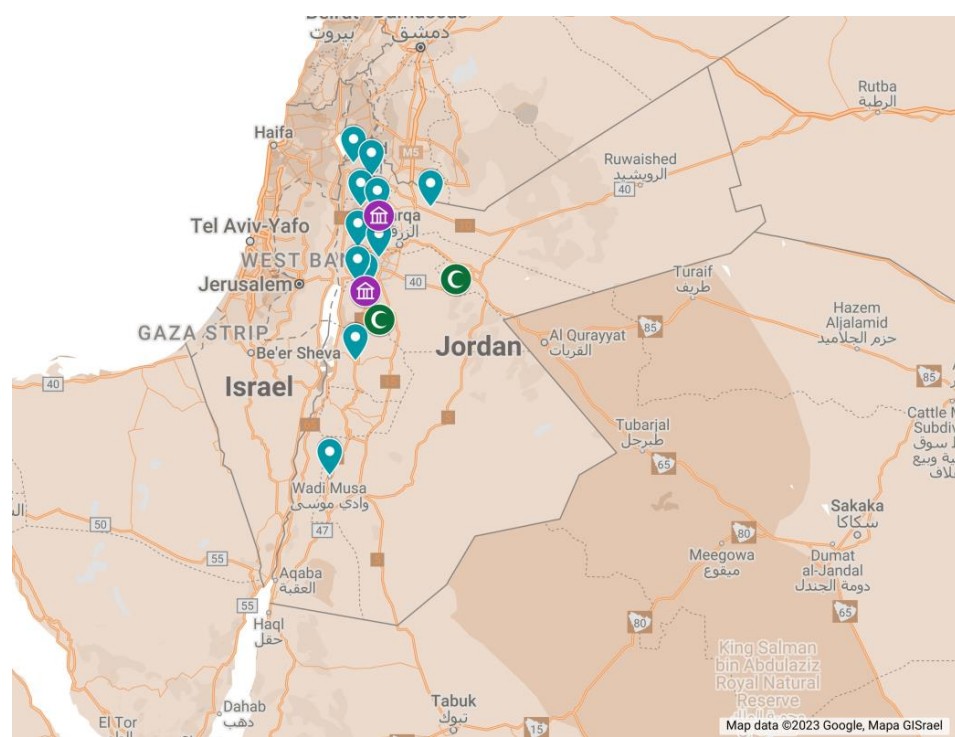

**Figure 1.** City with Classical Mosaic Pavements (Researcher, 2023) Source: Google map, March 2023; (https://www.google.com/maps/d/u/0/edit?mid=1gSl9lV4HoNUDgxjqJJtBRFEzJGUpn4 5Y&ll=31.462881293933165%2C36.03371170000002&z=7) accessed on 2 March 2023, modified by researchers. : Roman Mosaic; : Byzatione Mosaic; : Islamic Mosaic.

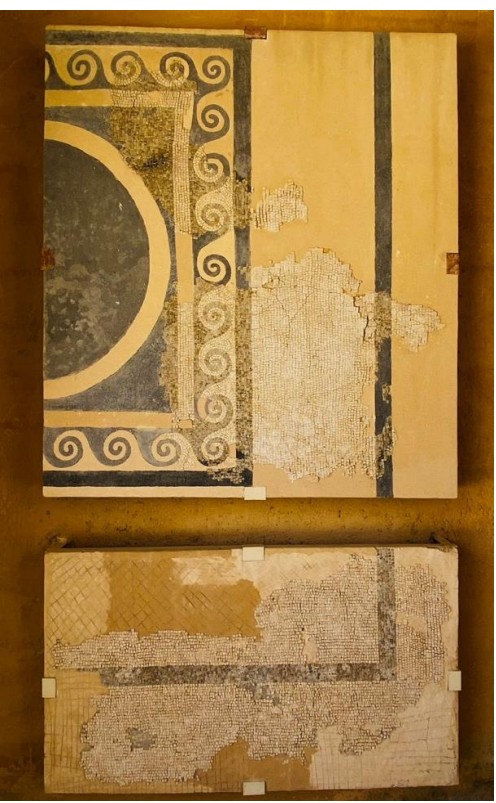

**Figure 2.** Mukawir Mosaic—The Madaba Archaeological Park (Researcher, 2023).

"Madaba wins world's mosaic city title" in 2016. The heritage of Madaba city is considered to be its mosaic heritage, which, despite its ancient history [45], also symbolizes its mosaic identity [46]. The city, moreover, its mosaic heritage, is still used today, and the Madaba Institute of Mosaic Art and Restoration [47] still considers Madaba to be the cradle of teaching the mosaic craft. The Madaba Institute of Mosaic Art and Restoration is a renowned institution located in Madaba, Jordan. It was created to promote and preserve the ancient traditions of mosaic art and restoration in the region.

In Jordanian society, Byzantine mosaic heritage is not limited to archaeological sites; it can be observed that copies of heritage mosaic appear in several contemporary Jordanian places, such as private residential places and public places (government, parks, malls, etc.) (Table 1), in the form of frames of diverse sizes, murals, and pavements, albeit being far from the current era. This phenomenon is described in this study as a way for community participation in heritage practice.

**Table 1.** The structure of the research: objectives, tools, and content.

| Objective | Tools | Content |
|---|---|---|
| Identify the level of Byzantine Place Attachment | Questionnaire | Information on the meaning, feelings, and identity of Byzantine mosaics |
| Educating The Community | Questionnaire | Information on the heritage Byzantine mosaic: - Awareness and knowledge - Government's efforts |
| Identify The Replicated Heritage Discourse | Questionnaire | Information on the replicated heritage discourse: - Identity—Authenticity—Prestige |

*2.2. Heritage and Heritage Discourse*

2.2.1. Heritage

The term "heritage" is defined by the Oxford English Dictionary as the environmental, cultural, and human inheritance transmitted down from the past. Heritage encompasses both natural and man-made landscapes, physical cultural forms (for example, music, literature, art, folklore, and monuments), intangible cultural aspects (for example, values and traditions, customs, spiritual beliefs, and language), as well as biometrics. Heritage is regarded as significant because it links people to the past and validates and replicates cultural identities [48].

The creation of dimensional value is sped up by procedures that are sparked by heritage, but it is also a part of well-being. The preservation of the asset's tangible and intangible values is a crucial basis for defining paths for economic progress, enhancing community well-being and quality of life from a sustainable perspective, and transmitting resources [49].

According to Ashworth and Tunbridge [22], heritage is created through interpretation, not only of what is interpreted but also of how and by whom it is interpreted. As a result, quite specific messages will be created regarding the significance and worth of heritage places and the history they represent. The aforementioned are referred to as Heritage Discourse.

2.2.2. Heritage Discourse

Discourse, according to Wetherell [50], is the "study of language use"(p. 3), an examination of how language is used "to do things"; however, it is not linguistically reducible (see the study from Taylor [51]). Hajer [52] defines the concept of discourse as a collection of particular ideas, concepts, and taxonomies produced, repeated, and transformed within a specific practice and used to give meaning to physical and social reality.

As a result, the compilation of thoughts, ideas, and classifications regarding heritage gives rise to various ways of "seeing" the social practice of managing "heritage" based on the positions of social actors [24]. The practice of heritage can be defined as the management and conservation protocols, procedures, and techniques used by heritage managers, archaeologists, museum curators, architects, and other experts [26].

Heritage discourse not only defines who has authority or "responsibilities," but it also allows the community to participate in heritage practice. In this context, some heritage studies have demonstrated a significant desire to identify and involve the community in the management of heritage, interpretation, and preservation efforts, which is commonly referred to as educating the community (see, for instance, [29,53]). Educating the community aims to spread the value and meaning of historical buildings and monuments, to guarantee increased conservation awareness and appreciation for the cultural heritage [26].

Regarding historical tourism, the perceptions and knowledge of individuals can be influenced by marketing efforts, government initiatives, and policies that are originally aimed at developing tourism [27]. Each of these strategies has an impact on the community's knowledge and participation in the practice of heritage and thus the consolidation of heritage. On the other hand, Byrne [54] argues that what makes heritage a permanent anchor is the conscious or unconscious belief in its authenticity.

### 2.3. Authenticity

"Authenticity" refers to something genuine, real, and true, rather than being fictional or imitative. The term has also been used in different fields of study such as the sociology of tourism, heritage conservation, and identity work [55,56].

Most scholars have agreed unanimously on separating authenticity into object-related authenticity (for example, art, artefacts, or buildings) [57–59] and activity-related authenticity (for example, thoughts and impressions generated by participating in tourism activities) (see [59,60]).

Object-related authenticity includes objectivist authenticity and constructivist authenticity. Objectivist authenticity refers to the credibility of assets used to verify their authenticities, such as cultural and historical sites, works of art, and artefacts, which are considered relatively fixed [58,60–66].

According to Wang [59], it is well known that local society emphasizes constructive authenticity based on cultural norms. The sense of authenticity could increase the level of place attachment [66].

### 2.4. Sense of Place and Attachment to Place

Sense of place refers to the sentimental connections and attachments that people form or feel to specific places and environments. Sometimes, the phrase "sense of place" is used to define the uniqueness or distinct character of certain places and regions. Sense of place can also be called the feeling people get from a place rather than what the place itself says. It is a feature that makes a place unique [6,66–68]. It is beyond a place's physical and sensory properties and may be inferred from the traits of the location and its inhabitants, their movements, everyday connections, and sentiments associated with the place [69]

Place attachment and sense of place, according to Williams and Vaske [70], can be practically equivalent in some fields. On the other hand, Trentelman [71] stated that place attachment refers only to a positive sense of place that occurs when a place is valued [8]. The concept of place attachment, which includes place identity, is generally acknowledged to be multifaceted [7,72–76] and place dependent [75,77,78].

Place identity is the relationship between a place and a person's identity [79,80]; this connection has both mental and emotive components, which is a significant part of one's overall sense of identity [81]. In addition, individuals can identify themselves through the cultural context of a unique place [82] and may cultivate and communicate their sense of identity [83,84], especially when a place is a place that gives a sense of distinctiveness or aids to differentiate a place from other places. Stokols and Shumaker [78] described

place dependence as the functional attachment of visitors to a specific area and their understanding of the distinctiveness of the place, which helps visitors accomplish their visiting objectives [85], where the place can be relied upon to construct and identify one's self-identification [86,87], and where the sense of place identity increases depending on the geographic and cultural composition of the place [18]. The power of the influence of attachment to heritage places and artefacts depends on the power granted to them in heritage discourse and on the way they are perceived and valued as items of belonging, identity, desire, status, authenticity, or other socio-cultural values [26].

### 2.5. Replicated Heritage

Generally, the importance of producing replicas lies in their cultural significance to society [81]. In other words, replicated heritage is accepted based on the discourse it carries, which is a result of the culture of the society (community education).

In this setting, heritage transforms into a cultural tool by which society expresses and fosters a sense of identity and belonging. For this reason, the "power of place" is invoked in its symbolic sense [26].

Holtorf [88] argued that replications are capable of creating "authenticity effects". According to Foster and Jones's [34] research, the replica must be accurate in terms of color, details, texture, or complementary materials, among other things, because the antique appearance and patina contribute to the authenticity [89]. Smith [26] noted that in the markets, replications of heritage are evaluated depending on the degree of 'authenticity' of materials used and the attention paid to replicating the object, where the most expensive version of the replicated object is very detailed and finely crafted. Furthermore, the social ranking of individuals is determined by the type of replications they purchase; so, replications are capable of creating "prestige effects".

### 3. The Theoretical Model

Figure 3's theoretical model explains the relationship between the structures associated with Jordanian society's mosaic heritage in terms of a logical flow. As evidenced by the literature (e.g., [6,66,68,90]), the sense of place is the emotional connections and attachments that people develop or feel in particular locations and surroundings; the positive sense of place is defined as "place attachment" [71]. The idea of place attachment, which includes two components, namely, place identification and place dependence, is used to measure this research's emphasis on the positive benefits of the mosaic heritage place.

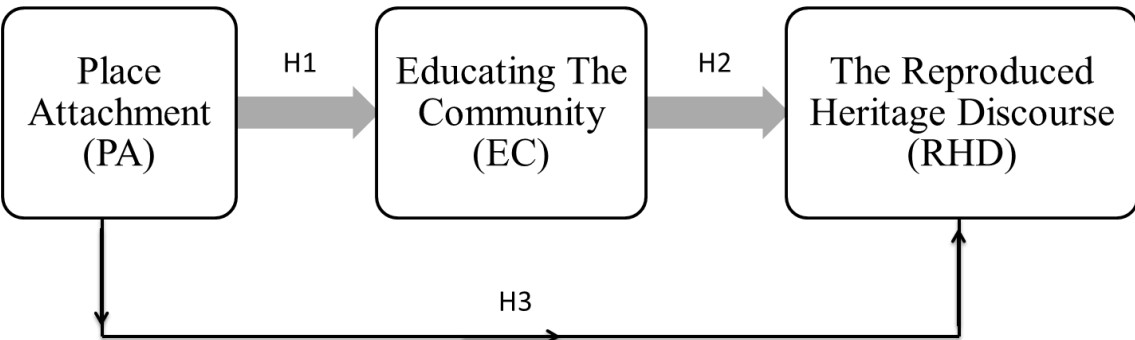

**Figure 3.** Theoretical Framework of the Study.

In addition, Scannell and Gifford [72] pointed out that one of the powers of place attachment is manifested through cognition, thereby realizing the value of the heritage place, preserving it, and educating the community about it. As such, the model proposes that attachment to Byzantine mosaic places has a positive impact on educating the community.

Furthermore, the literature suggests the value of replicas comes from their heritage discourse in society, which is created by educating the community, which essentially aims

to disseminate the value, significance, and meaning of historical buildings and monuments to ensure increased conservation awareness and appreciation for heritage, which leads to a different view of the social practice of managing heritage replicas [26,91,92]. Therefore, the model suggests that educating the community has a positive impact on the formation of the replicated heritage discourse and prompts replicated heritage consumption in contemporary places. Besides that, Smith [26] pointed out that the attachment to heritage place is a central concept in heritage discourse, which affects heritage consumption and acceptance. So, the model proposes that place attachment has a positive impact on the formation of the replicated heritage discourse, prompting replicated heritage consumption in contemporary places.

## 4. Method

There are several methods to define the concept of place attachment, each of which carries a slightly different theoretical implication. Place attachment refers to the emotional bonds that individuals form with various locations. Tradition divides the study of place attachment into two distinct schools of thought: the qualitative approach, which is based on geographic analyses of the sense of place, and the psychometric approach, which is founded on early community studies.

### 4.1. Research Design

To investigate the relationship between the attachments to the Madaba Archaeological Park, the community's level of awareness, and the replicated heritage discourse (replicated mosaics), structural equation modelling (SEM) was used to examine the relationship between these factors. Confirmatory factor analysis (CFA) was used to review the measurement model of the latent components and examine their construct validity and reliability. The structure of the research is summarized in Table 1.

### 4.2. Madaba City—The Research Location

Madaba is 33 km from the capital city of Jordan—Amman (Figure 4); Madaba was an important town at the beginning of the Christian era [41]. Imposing churches were built there, which have well-known mosaic pavements, such as those at St. George's Church, where the map of Madaba is located (the Holy Land), which dates back to the year 560 AD (Figure 5).

To preserve Madaba's rich heritage and make it accessible to visitors, Madaba Archaeological Park was established in 1991 by The Jordanian Ministry of Tourism and Antiquities, the American Center of Oriental Research (ACOR), and the United States Agency for International Development (USAID). Madaba Archaeological Park has Jordan's numerous most prominent mosaics [93], such as the Madaba Tree and the myth of Aphrodite and Adonis (Figure 6).

Furthermore, the Madaba Mosaic School was founded in 1992 next to the Madaba Archeological Park, which was transformed into the Madaba Institute for Mosaic Art and Restoration (MIMAR) in 2007, a non-profit government entity established to serve as a unique regional center of excellence for the conservation of Jordan's cultural and historical heritage mosaics. Throughout its history, the school and institute have received technical, administrative, and financial assistance from the government, NGOs, the Jordanian people, the Italian government, and the United States. The campus is located around a historic Roman road and situated next to the Madaba Archeological Park; this places MIMAR in a prime position for accessing around 400 mosaics sites in Madaba and throughout Jordan. The institute offers the only diploma program in the region specializing in the scientific methods of restoration and conservation, together with the mosaic art's creative components [45].

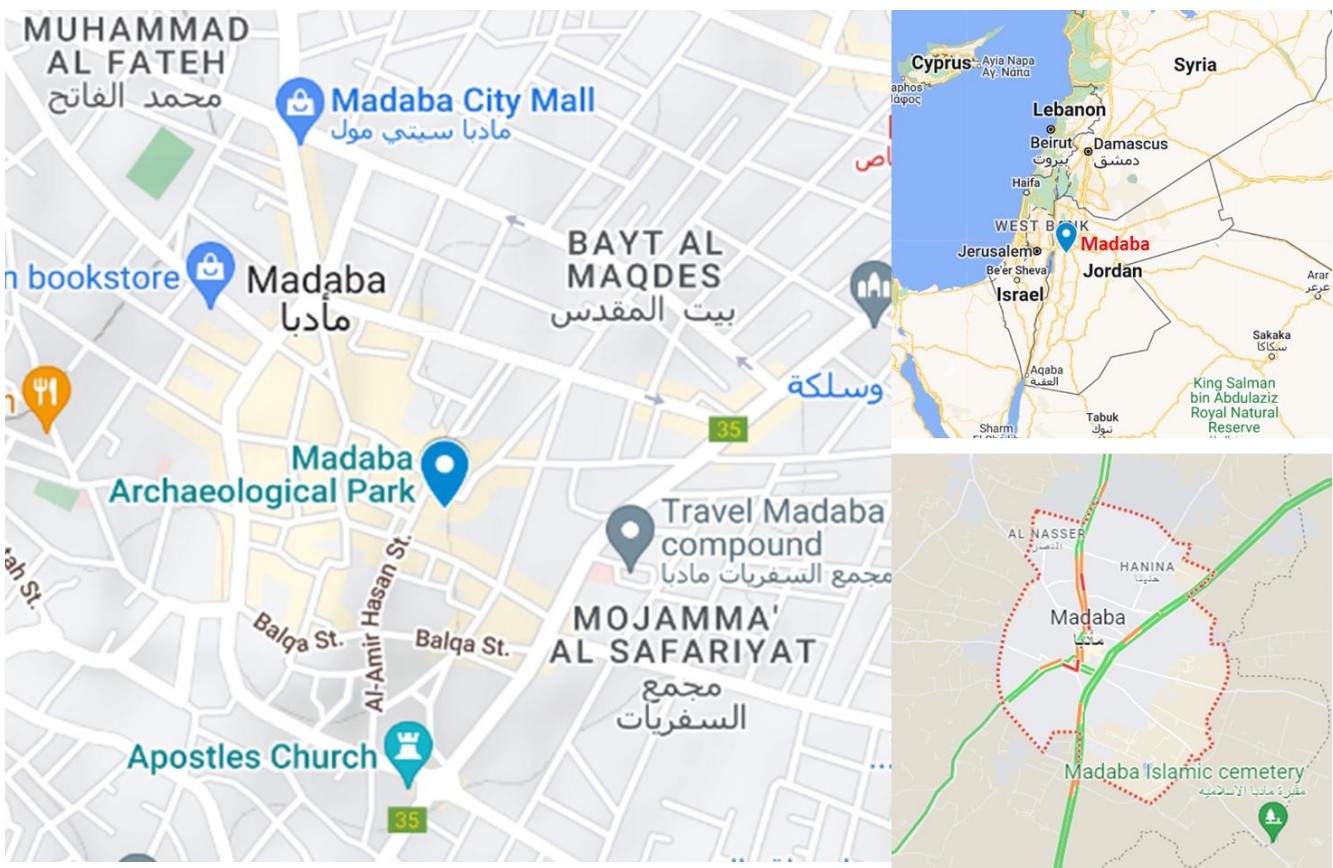

**Figure 4.** Map of Madaba with the Madaba Archaeological Park. Source: Google map, March 2023.

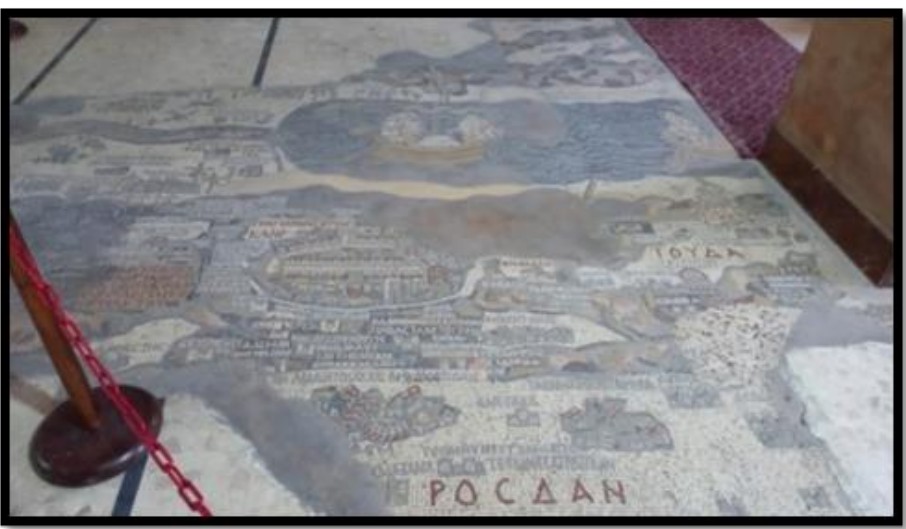

**Figure 5.** An original map of the Holy Land—church of Saint George (Researcher, 2023).

*4.3. Measures*

Based on previous studies, the model is designed to measure the influence of attachment to Byzantine mosaic places on the discourse of replicated mosaic heritage in contemporary places. The model included three constructs; the place attachment is the initial construct, and it consists of two parts: place identity and place dependence. Place identity included three items: "Byzantine mosaic means a lot to you", " I feel an attachment to the Byzantine mosaic sites", and "I feel a sense of belonging with Byzantine mosaic". Place dependence included three items: "I feel a sense of pride because of the historic

value of Byzantine mosaic sites at Madaba", "No other place can compare to the Madaba mosaic", and "The presence of Byzantine mosaic in a place makes me want to visit it". The second construct is that of educating the community, which included three items: "My awareness and concern about the issues related to classical mosaic are highly sufficient. (Ex: symbolism and myths.)", "The Byzantine mosaic is considered as mosaic heritage in the Jordanian community", and "The government has made efforts to sustain the mosaic craft to ensure that the mosaic heritage is preserved in the community". The third construct is the replicated heritage discourse, which included three items: "Adding heritage mosaic replication to the contemporary place gives it a special identity", "Adding heritage mosaic replication to the contemporary place gives a sense of authenticity", and "Adding heritage mosaic replication to the contemporary place gives it more prestige" (Table 2). All of the items were scored using a 5-point Likert scale, with 1 being the strongest disagreement and 5 being the strongest agreement.

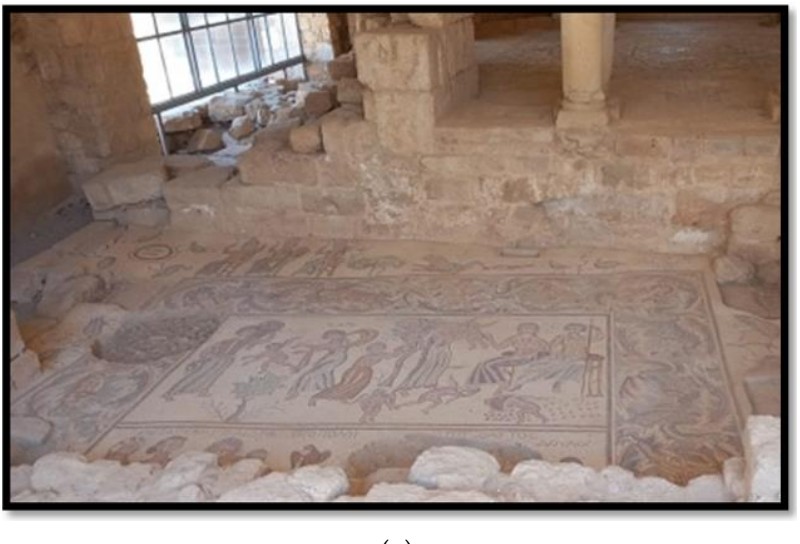
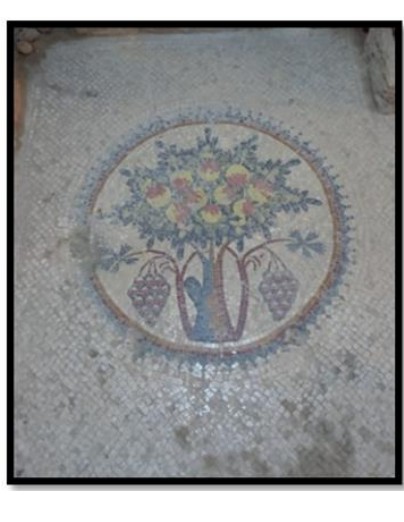

(**a**)  (**b**)

**Figure 6.** (**a**) The Myth of Aphrodite and Adonis—Virgin Mary Church; (**b**) The Madaba Tree—Church of St Elijah in Madaba (Researcher, 2023).

**Table 2.** Proposed measures for the study scale.

| Factors | Measures | Measures Used in Previous Studies | References |
|---|---|---|---|
| Place Attachment (PA) | Byzantine mosaic means a lot to you. | This trail means a lot to me | [94] |
| | I feel an attachment to the Byzantine mosaic sites. | I am very attached to X I am very attached to the Appalachian Trail. | [66] [94] |
| | I feel a sense of belonging with Byzantine mosaic. | I feel a strong sense of belonging to this National park | [78] |
| | No other place can compare to the Madaba mosaic. | No other place can compare to the Palace Museum | [62] |
| | I feel a sense of pride because of the historic value of Byzantine mosaic sites at Madaba. | | Exploratory study |
| | The presence of a Byzantine mosaic in a place makes me want to visit it. | | Exploratory study |

**Table 2.** *Cont.*

| Factors | Measures | Measures Used in Previous Studies | References |
|---|---|---|---|
| Educating the Community (EC) | My awareness and concern about the issues related to classical mosaics are highly sufficient. (Ex: symbolism and myths.) | | Exploratory study |
| | The Byzantine mosaic is considered a mosaic heritage in the Jordanian community. | | Exploratory study |
| | The government has made efforts to sustain the mosaic craft to ensure that the mosaic heritage is preserved in the community | | Exploratory study |
| The Replicated Heritage Discourse (RHD) | Adding heritage mosaic replication to the contemporary place gives it more prestige. | | Exploratory study |
| | Adding heritage mosaic replication to the contemporary place gives it a special identity. | | Exploratory study |
| | Adding heritage mosaic replication to the contemporary place gives a sense of authenticity. | | Exploratory study |

*4.4. Data Analysis*

Numerous statistical techniques were employed to evaluate the data from the present study. Initially, frequencies and percentages were utilized to characterize the research participants' demographic data. Next, mean values were calculated to provide insight into each item. Subsequently, Smart PLS version 4 was utilized to test the research hypotheses through path analysis.

**5. Results**

*5.1. Study Sample*

A total of 128 participants participated in the study, with 51.6% (*n* = 66) identified as female and 48.4% (*n* = 62) identified as male. The majority of participants fell into the 31–46 age range (43.0%, *n* = 55), with 27.3% (*n* = 35) in the 15–30 age range, 26.6% (*n* = 34) in the 47–62 age range, and 3.1% (*n* = 4) falling into the "Other" category. In terms of education, the largest group of participants were postgraduate students, comprising 66.9% (*n* = 85) of the sample. A smaller number of participants had completed undergraduate degrees (26.0%, *n* = 33), while only a few participants had completed elementary or secondary education (4.7%, *n* = 6 and 1.6%, *n* = 2, respectively). Finally, in terms of occupation, the largest group of participants were employed (54.7%, *n* = 70), followed by unemployed (23.4%, *n* = 30), students (11.7%, *n* = 15), and retired individuals (10.2%, *n* = 13). These demographic characteristics provide an important context for understanding the attitudes and perceptions of the participants towards classical mosaic replication (Table 3).

*5.2. Questionnaire Result Description*

The results presented in Table 4 all items had high mean values, indicating a strong attachment to classical mosaic designs among participants. The item "Byzantine mosaic means a lot to you" had a mean of 4.42 (SD = 0.79), and "I feel an attachment to the Byzantine mosaic designs" had a mean of 4.44 (SD = 0.77). I feel a sense of belonging with Byzantine mosaic was also high, with a mean of 3.74 (SD = 1.04). Participants demonstrated a sense of pride due to the historic value of a place, which was not comparable

to other places, with a mean of 4.43 (SD = 0.878). In terms of knowledge, "No other place can compare to the Madaba mosaic" had a mean of 4.22 (SD = 0.963), and the presence of classical music in a place making participants want to visit it had a mean of 4.30 (SD = 0.845). Overall, the results indicated a high level of place attachment among the study participants, with a combined mean of 4.26 (SD = 0.88).

**Table 3.** Percentage and percentage of the sample characteristics.

| Variable | Category | N | % |
|---|---|---|---|
| Gender | Female | 66 | 51.6 % |
| | Male | 62 | 48.4 % |
| Age | 15–30 | 35 | 27.3 % |
| | 31–46 | 55 | 43.0 % |
| | 47–62 | 34 | 26.6 % |
| | Other | 4 | 3.1 % |
| Education | Elementary Education | 6 | 4.7 % |
| | Postgraduate | 85 | 66.9 % |
| | Secondary Education | 2 | 1.6 % |
| | Undergraduate | 34 | 26.8 % |
| Occupation | Employed | 70 | 54.7 % |
| | Retired | 13 | 10.2 % |
| | Student | 15 | 11.7 % |
| | Unemployed | 30 | 23.4 % |

**Table 4.** The mean and standard deviation for the items of the study variables.

| Factors | Items | Mean | SD |
|---|---|---|---|
| Place Attachment (PA) | Byzantine mosaic means a lot to you. | 4.42 | 0.79 |
| | I feel an attachment to the Byzantine mosaic sites. | 4.44 | 0.77 |
| | I feel a sense of belonging with Byzantine mosaic. | 3.74 | 1.04 |
| | No other place can compare to the Madaba mosaic. | 4.22 | 0.963 |
| | I feel a sense of pride because of the historic value of Byzantine mosaic sites at Madaba. | 4.43 | 0.878 |
| | The presence of a Byzantine mosaic in a place makes me want to visit it. | 4.30 | 0.845 |
| | Overall | 4.26 | 0.88 |
| Educating the Community (EC) | My awareness and concern about the issues related to classical mosaics are highly sufficient. (Ex: symbolism and myths.) | 3.97 | 0.941 |
| | The Byzantine mosaic is considered a mosaic heritage in the Jordanian community. | 4.12 | 0.885 |
| | The government has made efforts to sustain the mosaic craft to ensure that the mosaic heritage is preserved in the community. | 4.51 | 0.789 |
| | Overall | 4.20 | 0.87 |
| The Replicated Heritage Discourse (RHD) | Adding heritage mosaic replication to the contemporary place gives it more prestige. | 4.54 | 0.626 |
| | Adding heritage mosaic replication to the contemporary place gives it a special identity. | 4.55 | 0.74 |
| | Adding heritage mosaic replication to the contemporary place gives a sense of authenticity. | 4.16 | 0.83 |
| | Overall | 4.42 | 0.73 |

Based on the results presented in the table, it can be inferred that the participants have positive attitudes towards educating the community and its preservation. The mean score for the first item was 3.97 (SD = 0.941), suggesting that participants felt they had a moderate level of awareness and concern about the issues related to Byzantine mosaics, such as symbolism and myths. The mean score for the second item was 4.12 (SD = 0.885), indicating that participants recognized the historical and cultural significance of Byzantine mosaics in the area.

The mean score for the third item was the highest among the three, at 4.51 (SD = 0.789), indicating that participants strongly believed that the government has made efforts to sustain the mosaic craft to ensure that the mosaic heritage is preserved in the community. The overall mean score for educating the community was 4.20 (SD = 0.87), indicating that the participants had a generally positive attitude towards Byzantine mosaic culture and its preservation.

These findings suggest that there is a need for continued efforts to preserve classical mosaic culture and its heritage among the people, as it is highly valued by the participants.

The results presented in the table indicate that participants expressed a high degree of the replicated heritage discourse with places that have Byzantine mosaic replication. The mean scores for the first two items were both above 4, with means of 4.54 (SD = 0.626) and 4.55 (SD = 0.74), respectively. These findings suggest that adding Byzantine mosaic replication to a place enhances its prestige and gives it a special identity.

The mean score for the third item was lower, at 4.16 (SD = 0.83), but still relatively high, indicating that participants generally feel a sense of the replicated heritage discourse when visiting places that are covered with mosaic replication. The overall mean score for the replicated heritage discourse was 4.42 (SD = 0.73), indicating that participants were highly satisfied with places that have Byzantine mosaic replication.

The results suggest that investing in classical mosaic replication can contribute to visitors' overall the replicated heritage discourse.

### 5.3. Measurement Model Validity—Confirmatory Factorial Analysis (CFA)

Confirmatory factor analysis (CFA) is a statistical technique used to validate hypothesized theoretical constructs or factors by determining how closely a group of observable variables resembles the theoretical concept they are intended to represent. CFA aims to identify a set of items that can explain all the constructs or factors hypothesized in the study through path loading. The CFA results can be combined with construct validity tests to raise the standards of the measurements. In this study, the constructed structures were specified according to a thorough exploratory investigation, and each structure was refined and confirmed using CFA before building the measurement model. The measuring model specifies the relationships between observed and latent variables or hypothetical constructs, and reliability and validity issues were identified before the structural equation model was fitted (Webster and Fisher, 2001).

Table 5 shows three factors, namely, place attachment (PA), classical mosaic culture (EC), and replicated heritage discourse (RHD). The loading values presented in the table show the factor loadings associated with each research item.

For the place attachment factor, items PA1, PA2, PA3, PA4, PA5, and PA6 have factor loadings of 0.762, 0.822, and 0.906, respectively. For educating the community factor, all three items (EC1, EC2, and EC3) have high factor loadings, with values of 0.841, 0.861, and 0.885, respectively. For the replicated heritage discourse factor, items RHD 1, RHD 2, and RHD 3 have factor loadings of 0.805, 0.844, and 0.862, respectively.

The high factor loadings indicate that each item has a strong relationship with its corresponding factor, suggesting that the items are measuring the intended constructs. The factor analysis's findings indicate that the three factors identified in the study (place attachment, educating the community, and replicated heritage discourse) are distinct and can be used to explore different aspects of visitors' experiences with Byzantine mosaic sites.

**Table 5.** Loading value for the CFA for study variables.

| Factor/Items | PA | EC | RHD |
|:---:|:---:|:---:|:---:|
| EC1 | | 0.841 | |
| EC2 | | 0.861 | |
| EC3 | | 0.885 | |
| PA1 | 0.877 | | |
| PA2 | 0.822 | | |
| PA3 | 0.869 | | |
| PA4 | 0.762 | | |
| PA5 | 0.887 | | |
| PA6 | 0.906 | | |
| RHD1 | | | 0.805 |
| RHD2 | | | 0.844 |
| RHD3 | | | 0.862 |

Table 6 presents the reliability and validity measures for the three factors identified in the study, namely, place attachment (PA), educating the community (EC), and replicated heritage discourse (RHD); the measures include Cronbach's alpha, composite reliability (rho_a), composite reliability (rho_c), and average variance extracted (AVE).

**Table 6.** AVE and Reliability for Study Variables.

| | Cronbach's Alpha | Composite Reliability (rho_a) | Composite Reliability (rho_c) | The Average Variance Extracted (AVE) |
|:---:|:---:|:---:|:---:|:---:|
| EC | 0.828 | 0.829 | 0.897 | 0.744 |
| RHD | 0.789 | 0.807 | 0.876 | 0.701 |
| PA | 0.818 | 0.826 | 0.891 | 0.733 |

The Cronbach's alpha values for the three factors range from 0.789 to 0.828, which indicates well-to-excellent internal consistency. The composite reliability (rho_a) values range from 0.807 to 0.829, and composite reliability (rho_c) values range from 0.876 to 0.897, which are also good indicators of reliability.

The average variance extracted (AVE) values range from 0.701 to 0.744, which indicates that the measures are valid and are capturing a substantial amount of variance in the constructs being measured.

*5.4. Model Fitness*

Table 7 presents fit indices for a saturated model and an estimated model in the study. The fit indices include standardized root mean square residual (SRMR), root mean square error of approximation (d_ULS), root mean square residual (d_G), Chi-square, and normed fit index (NFI).

**Table 7.** Study model fitness.

| | Saturated Model | Estimated Model |
|:---:|:---:|:---:|
| SUMMER | 0.07 | 0.078 |
| d_ULS | 0.385 | 0.477 |
| d_G | 0.284 | 0.305 |
| Chi-square | 204.405 | 214.242 |
| NFI | 0.806 | 0.796 |

The estimated model has a slightly worse fit than the saturated model, with higher values for SRMR, d_ULS, d_G, and Chi-square, and a lower value for NFI. The computed model's fit indices, however, remain within a respectable range, suggesting a good match.

These results suggest that the estimated model is a good representation of the data and that the model can provide valuable insights into the relationships between the variables being studied.

### 5.5. Results of the Path Model

- Hypothesis 1: Place attachment has a positive impact on educating the community.

To test this hypothesis, a path analysis was performed to examine the direct effects of place attachment on mosaic culture (educating the community).

Table 8 and Figure 7 present the outcomes of the path analysis. The analysis revealed a statistically significant and robust correlation between place attachment (PA) and mosaic culture (EC). The path coefficient from PA to EC was 0.83, indicating a strong relationship between the two variables. The t-statistic was 22.38, and the *p*-value was 0.00, providing further evidence of the relationship's strength.

**Table 8.** The results of the model (standardized regression weights).

| Hypothesized Paths | β | S.E. | z-Value | *p*-Value | Hypotheses |
|---|---|---|---|---|---|
| H1: PA →EC | 0.83 | 0.04 | 22.38 | 0.00 | Supported |
| H2: EC →RHD | 0.44 | 0.14 | 3.22 | 0.00 | Supported |
| H3: PA →RHD | 0.40 | 0.14 | 2.89 | 0.00 | Supported |

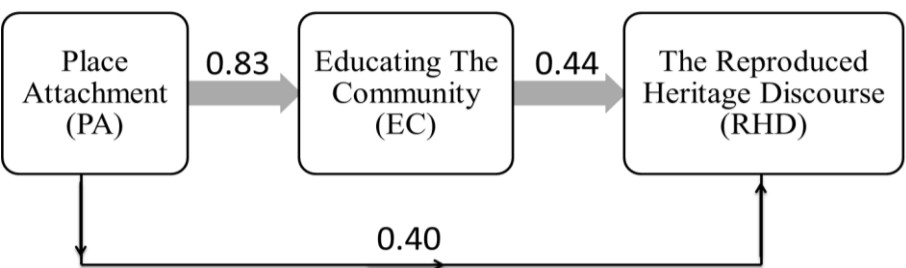

**Figure 7.** The results of the path model are based on standardized regression weights.

Higher levels of place attachment are associated with greater community education, according to these results. In other words, the greater a person's emotive connection to a place, the more likely they are to be interested in and engaged with that place's cultural heritage. This has significant implications for the preservation of cultural heritage sites and heritage discourse. Communities may be more likely to value and protect their cultural heritage and take an active role in educating others about it if they cultivate a sense of place attachment.

The results of this study provide substantial support for the hypothesis that place attachment has a positive effect on community education. The findings emphasize the importance of understanding the affective connections people have with locations and the role these connections may play in influencing attitudes toward cultural heritage.

- Hypothesis 2: Educating the community has a positive impact on the formation of the replicated heritage discourse in contemporary places.

To investigate this hypothesis, the researchers conducted a path analysis to determine the direct relationship between mosaic culture (community education) and replicated heritage discourse. The results of the path analysis indicate that EC and RHD have a strong and statistically significant relationship. The path from EC to RHD had a path coefficient of 0.44, resulting in a high t statistic of 3.20 and a *p*-value of 0.00.

Higher levels of EC are associated with higher levels of RHD, indicating a strong and positive correlation between these two variables. This finding supports the hypothesis that community education has a positive effect on the reproduction of heritage discourse in contemporary locations.

In other words, the greater a community's knowledge of the heritage and culture of a place, the greater their likelihood of replicating and preserving that heritage in contemporary places. This information can be used to inform the development of policies and interventions intended to promote the preservation and replication of heritage in contemporary locations.

- Hypothesis 3: Place attachment has a positive impact on the formation of the replicated heritage discourse, prompting replicated heritage consumption in contemporary places.

To test this hypothesis, a path analysis was conducted to investigate the direct effects of place attachment on replicated heritage discourse. The path analysis revealed a significant and robust relationship between place attachment and replicated heritage discourse. The place attachment to replicated heritage discourse path had a path coefficient of 0.40, resulting in a high t-statistic of 2.89 and a *p*-value of 0.00.

The results indicate a positive correlation between place attachment and the development of replicated heritage discourse. This suggests that as people's attachment to a place increases, they are more likely to engage in the consumption of replicated heritage in modern locations. The level of replicated heritage discourse and consumption increases proportionally with the level of place attachment.

These findings contribute to the comprehension of the function of place attachment in promoting the consumption of replicated cultural heritage. By understanding how place attachment influences replicated heritage consumption, policymakers and heritage administrators can develop strategies to promote and enhance place attachment to increase heritage consumption and maintain the site's vitality.

## 6. Discussion and Conclusions

This study aimed to examine the role of attachment to heritage places in shaping the discourse of replicating heritage in modern architectural environments, specifically Byzantine mosaics in Jordan. The survey found that participants had a strong emotional connection to mosaic heritage places (Madaba Archaeological Park) and that this connection was significantly influenced by the historical significance and distinctive qualities of Byzantine mosaics. Byzantine mosaics are recognized as a national identity in Jordanian society [44], where people may define their identity via the cultural environment of distinctive places, such as Madaba Archaeological Park. This is especially true when a location stands out from other locations because of its monumentality, uniqueness, or aesthetic appeal, which is in line with other studies [26].

The survey also discovered that although participants valued the Byzantine mosaic's place in cultural history, they did not feel a strong feeling of belonging to it. These findings show the intricate connection between cultural heritage, people's sense of identity, and their attachment to place.

There are important insights into how reliance on knowledge and familiarity with and appreciation of heritage influence the formation of a sense of identity, resulting in an attachment to Madaba Archaeological Park, which is seen as a positive sense of place [71]. As such, the consistency and low variability of responses suggest that these factors are important in attachment formation.

Efforts to educate the community about the Byzantine mosaic heritage and preserve it has been successful, one example being the Madaba Institute for Mosaic Art and Restoration, a government-owned, non-profit entity established to promote and preserve the ancient tradition of mosaic art and restoration in the region, which also offers basic and advanced courses in traditional mosaic techniques, as well as educating the community [45]. As a

result of these efforts, Madaba City is still regarded as the cradle of teaching mosaic craft in the region.

There are significant insights into the strategy of community education: first, promoting and preserving mosaic sites by establishing Madaba Archaeological Park and reviving the mosaic craft; second, highlighting the historical significance and distinctive qualities of the mosaics in the society; third, considering the mosaic as a national identity in society. As such, the consistency and low variability of responses suggest that these factors are important in educating the community.

In addition, the survey shows that mosaic heritage in Jordanian contemporary places is a cultural tool used to express and create a sense of identity, while heritage mosaic replications are capable of creating "authenticity effects" [33,34] by invoking the symbolic power of place. Additionally, the closer a replica is to the original, the greater the impression of authenticity and the more valuable the replica, which creates a sense of prestige in the place; these results are in line with other studies [26,33,34,91].

In other words, the acceptance of the placement of replicated heritage in modern architectural environments is largely dependent on the discourse surrounding the replicated heritage, which depends on cultural norms of Jordanian society (identity, authenticity, prestige), As such, the consistency and low variability of responses suggest that these factors are important in the replicated heritage discourse formation in contemporary places.

The hypotheses testing results revealed a strong and positive association between place attachment and educating the community (mosaic culture). Higher levels of place attachment were discovered to be linked to higher levels of community education. These findings suggest that individuals who have a strong attachment to a place are more likely to value and appreciate the cultural heritage of that place, specifically, in this case, Byzantine mosaics. This highlights the importance of fostering a sense of attachment to places to promote and preserve heritage [26].

Additionally, the study found that there is a strong and positive relationship between educating the community and replicated heritage discourse. Higher levels of educating the community were associated with higher levels of replicated heritage discourse, indicating that promoting education about heritage in communities may be an effective strategy for increasing discourse on and awareness of heritage. How heritage practices (such as educating the community) are carried out affects the kinds of social or cultural meanings that are formed, accepted, and communicated through discourse. These findings highlight the importance of supporting community education programs that promote heritage awareness and neighborhood customs including exhibiting replica mosaics in contemporary contexts and promoting dialogue about replicating heritage.

The indirect effect of place attachment on the replicated heritage discourse through community education is significant. This means that the relationship between park attachment and the replicated heritage discourse can be partially explained by the mediating role of social education. A more thorough and exact understanding of park attachment can result in heritage practices that better mimic the placement and reception of mosaics in contemporary places. The study underlines how important it is to foster a feeling of place attachment and cultural heritage to increase heritage appreciation and preservation. Educating the community about heritage and the government's efforts to preserve it can be successful. Furthermore, replica heritage can be valuable in modern architectural environments if society allows it.

In conclusion, the research found that individuals in Jordan have a strong attachment to Byzantine mosaics, specifically at Madaba Archaeological Park, where the study was conducted, which is manifested through realizing the historical value and unique features of mosaic heritage at Madaba, which are identified as part of the national identity. Because of the mosaic heritage's significance to Jordanian society, a special discourse on the replicated heritage in modern Jordanian places has been formed, based on cultural norms of society.

Overall, the popularity of heritage replications in contemporary places "relies on our growing historical interest", "Intellectually, at least, we inhabit a conceptual world

defined by historicism—the belief that knowing one's history brings value and interest to virtually everything." [95] (p. 38). So, the attachment, knowledge, and awareness of heritage places contribute to shaping the discourse surrounding the copied heritage in modern architectural environments.

### 7. Limitations and Suggestions

There are a few restrictions on the study that need to be mentioned. Firstly, the study only examines the Byzantine mosaic heritage, and each type of heritage has its own distinct and particular characteristics, so the results' generalizability is constrained. Therefore, the universality of the results of this study should be further tested, and future studies should also test the universality of the proposed model for other heritage types.

Secondly, this study only examined the role of attachment to Byzantine mosaic heritage places in shaping the discourse of replicating heritage in modern architectural environments, specifically Byzantine mosaics in Madaba, Jordan. However, visitors to other regional or national heritage sites may have a different perspective on mosaic heritage and the discourse of heritage replication in the modern built environment. Future research should thus be extended to other areas or nations with caution and should take into account variations in visitor perceptions caused by the existence of cultural or regional difficulties.

For future studies, the researchers recommend that the scale applies to a wider sample of different types of heritage in other regions. Furthermore, the researchers suggest the development of scales for RHD which are more comprehensive and accurate by including more replicable discourse scales.

**Author Contributions:** Conceptualization H.J.A., Z.O. and H.A.U.; methodology, H.J.A., Z.O. and H.A.U.; software, H.J.A.; validation, H.J.A., Z.O. and H.A.U.; formal analysis, H.J.A.; investigation, H.J.A.; resources, H.J.A.; data curation, H.J.A.; writing—original draft preparation, H.J.A.; writing—review and editing, Z.O. and H.A.U.; visualization, H.J.A.; supervision, H.A.U. and Z.O.; All authors have read and agreed to the published version of the manuscript.

**Funding:** This research received no external funding.

**Institutional Review Board Statement:** The study was conducted following the Declaration of Helsinki and approved by the Ethics Committee of Near East University (YDÜ/FB/2019/76–26.11.2019). The study was conducted following the Declaration of Helsinki and approved by the Institutional Review Board of the Madaba Institute of Mosaic Art and Restoration (R/212/1/2023–10/1/2023). All authors have read and agreed to the published version of the manuscript.

**Informed Consent Statement:** Informed consent was obtained from all subjects involved in the study.

**Data Availability Statement:** All data are available publicly, as explained in the full article.

**Conflicts of Interest:** The authors declare no conflict of interest.

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
