# Peer review of "The Influence of Place Attachment on Heritage Discourse in Contemporary Places: A Case Study of Jordanian Byzantine Mosaics"

_sustainability, doi:10.3390/su15108395_

Round 1
Reviewer 1 Report
The paper is well structured, the title is adequate and refers updated bibliography.
Review by English native is recommended. There are inappropriate commas and some improvement can be made in order to simplify ideas/sentences.
Review by English native is recommended. There are inappropriate commas and some improvement can be made in order to simplify ideas/sentences.
Author Response
Dear reviewer,
The comments have been addressed.
Thank you
Reviewer 2 Report
Introduction:
- It is basically a literature section, which I suggest moving it to the appropriate heading. The authors need to introduce the subject in general with social, political and cultural influence, and set some goals and objectives followed by a general overview of what they are trying to accomplish in this study. Also, a state of art is needed.
Methodology:
- The study focused heavily on numerical values even though the title suggests place attachment. Such a subject should focus on human behavior and how architecture and sculpture reflect tangible (the mosaic element) as intangible meanings (through art, history and culture). In other words, how such an architectural element (mosaic) is used in place to express user attachment to the place? To answer such a question, the authors needs to understand and examine how users behave in the place.
- That does not mean the methods deployed is incorrect; it is an interesting and different approach. However, it needs to be supported by exploring the behavior of the users and how they are reflecting to such an element. To be able to do so, I suggest reviewing the literature again and understanding how pioneer scholars explored the subject of place attachment. You will need some architectural analysis to justify your study output, and more importantly you will need an ethnographic method to reveal hidden and embedded meaning and attachments:
Amos Rapoport. (1969). House form and culture
Edward Relph. (1976). Place and Placelessness
Lewicka, M. (2011). Place attachment: How far have we come in the last 40 years?.
Altman, I., & Low, S. M. (2012). Place attachment
- Another issue is that the questionnaires are not related to place attachment theory and concepts; they are more leaned towards explanatory questions that seek to understand do users have familiarity and background of the element they are viewing in the place. The interview method needs to be deployed and questions about opinion, history, feelings, and emotions are necessary to understand the attachment processes.
General:
- The authors stated, "replicating heritage in modern architectural environments, specifically 513 Byzantine mosaics in Jordan." However, the study title indicates that the authors will explore the study from a place, architectural, behavioral, and anthropologist perspective. Therefore, the title is misleading and needs a rewrite.
- As a result of the previous concerns, I suggest removing the place attachment from the title as it requires different sets of objectives and methods to reach reasonable outcomes. Otherwise, the authors need to restructure the manuscript.
Author Response
Dear reviewer,
* The Introduction has been improved according to the feedback.
*The updates hypothesis includes more explanation.
* The method tries to explain the relation of humans and space and places; the approach of the method stays the way it was to keep the structure of the article and the guidelines of the thesis which is related to this article.
* The questionnaire hasn't been updated because changing the questionnaire will affect the scope of the study and leads to another perspective.
* About the interview, there is another part that is not included in the article (an interview with mosaic designers).
Thank you

Reviewer 3 Report
I read with great interest the article “The Influence of Place Attachment on Heritage Discourse in Contemporary Places: Jordanian Byzantine Mosaics Case Study”.
The subject matter is very interesting and has been tackled with a well-conducted and methodologically well-structured scientific approach; the conclusions are well argued and also contain interesting points for reflection as well as guidelines for future research developments. The bibliography is extensive and pertinent.
I would only propose a limited integration that could improve understanding; in detail, in point 5.5. “Results of the Path Model”, I would suggest to better discuss how the hypotheses introduced were analyzed and developed.
Author Response
Dear reviewer,
The updates hypothesis includes more explanation.
Thank you
Round 2
Reviewer 2 Report
Even though the revised parts are less than requested, the authors tried to accommodate as much as possible of the given comments to enhance the manuscript's quality. Therefore, based on the final approaches and outcomes and the new additions to the discussion of the study, I believe it is suitable for publication.